

# Duality in quantum transport models

**Rouven Frassek**[1⋆], **Cristian Giardinà**[2†] **and Jorge Kurchan**[1‡]

**1** Laboratoire de Physique de l'École Normale Supérieure, ENS, Université PSL, CNRS,
Sorbonne Université, Université de Paris, 75005 Paris, France
**2** University of Modena and Reggio Emilia, FIM, Via G. Campi 213/b, 41125 Modena, Italy

⋆ rouven.frassek@phys.ens.fr, † cristian.giardina@unimore.it,
‡ kurchan.jorge@gmail.com

## Abstract

We develop the 'duality approach', that has been extensively studied for classical models of transport, for quantum systems in contact with a thermal 'Lindbladian' bath. The method provides (a) a mapping of the original model to a simpler one, containing only a few particles and (b) shows that any dynamic process of this kind with generic baths may be mapped onto one with equilibrium baths. We exemplify this through the study of a particular model: the quantum symmetric exclusion process introduced in [1]. As in the classical case, the whole construction becomes intelligible by considering the dynamical symmetries of the problem.


# 1   Introduction

A central role in non-equilibrium statistical mechanics of classical systems is played by stochastic processes. For instance, in the study of mass transport associated to a non-equilibrium steady state with non-zero current, a pivotal role has been played by the *simple symmetric exclusion process* (SSEP), that has been the subject of intensive investigations since its introduction. In the study of this process emerge properties that are believed to be universal signatures of a non-equilibrium stationary states, such as long-range correlations (in turn the source of non-local large deviation functionals for the density). Similarly, non-gaussian fluctuations are observed in the asymmetric version of the process (related to the Kardar-Parisi-Zhang universality class). In both settings several tools from the theory of Markov processes have been used. We shall focus here on one such a tool, which is known in the probabilistic literature as *duality*.

Dual processes were introduced in the realm of interacting particle systems at the early days of the field by Spitzer and Liggett [2, 3]. For instance, the symmetric exclusion process on the lattice turns out to be *self-dual*, and this property has been heavily used and fundamental to develop the ergodic theory of exclusion process in infinite volume. In the non-equilibrium set-up, dual processes are also useful to study the stationary measure, which is necessarily non-reversible to sustain a current. Again, the simplest example is the open exclusion process on a chain, which is coupled at its ends to reservoirs which inject and remove particles at different rates. Here the dual process simplifies the analysis by *transforming the reservoirs into absorbing boundaries* [4]. In doing so, the study of correlation functions in the open system is reduced to following the dynamics of a few dual particles that are eventually absorbed in the boundaries.

Dual counterparts of the open exclusion process played a crucial role in the construction of the so-called *hydrodynamic limit* [5], i.e. a macroscopic theory described by partial differential equations. This turned out to be true for a large class of diffusive systems (Kipnis-Marchioro-Presutti process, symmetric inclusion process, Brownian energy process [6–11]) related to the study of Fourier's law of heat conduction. Other simplifications due to duality occurred in the study of asymmetric exclusion process on the infinite line [12–15], directly related to the height profile of interface growth models. There, duality helps the study of *fluctuations* around the hydrodynamic limit: the evolution of one dual particle is related to the microscopic version of the Cole/Hopf transform that maps the non-linear and ill-posed KPZ equation to the linear stochastic heat equation [15, 16].

Duality has also another surprising consequence. Years ago, Tailleur et al. [17] were puzzled by a crucial step associated with the solution of Bertini et al [18] of the *large deviations* (around the hydrodynamic limit) of a family of transport models: the fact that the explicit construction of a trajectory with time-reversed extremes was possible even out of equilibrium. The solution of the puzzle was that these models revert, via a non-local transformation, into problems with detailed balance. In a recent paper [19], we have shown that this is a general consequence of duality, to be expected if (and probably only if) some form of duality is present.

It is then natural to ask if duality may be introduced, and if so, which (if any) of the simplifications obtained by such a construction survive in a *quantum stochastic system* [1, 20–24]. This is precisely the question addressed in this paper. At first sight, the two problems may seem completely different, as the replacement of the Markov evolution with a quantum semigroup substantially changes the averaging procedure: quantum averages are quadratic in the wavefunction, whereas the probability density appears linearly in the averages of classical stochastic systems. However, as we shall further discuss below, there is a key argument that brings the two problems in close contact: in both settings, classical and quantum, the evolution

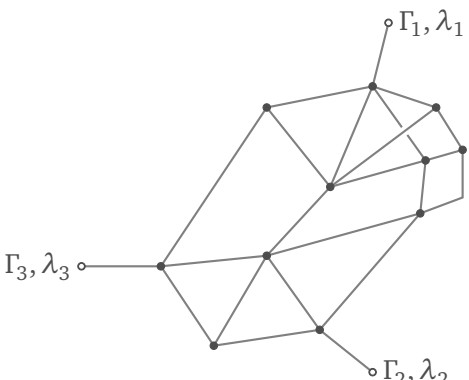

Figure 1: Schematic picture of a transport model on a graph. A non-equilibrium stationary state is attained as a consequence of interactions with sources at different parameters. The example shows three 'baths' with couplings $\Gamma_i$ and average densities of particles $\lambda_i$, with $i = 1, 2, 3$.

can be described in algebraic terms using a Lie algebra. For classical systems this was remarked years ago in a pioneering paper by Schütz and Sandow [25] and then further extended in [26]. Here we show, by focusing on a specific example, that quantum stochastic systems admits the same algebraic description, provided one moves to *superoperators*. This allows to repeat the pattern that lead to the formulation of duality in Markov systems.

The example we shall discuss is a system of free fermions with a noisy dynamics which has been extensively studied in [1, 23, 24]. The system plays a paradigmatic role for quantum stochastic evolutions, similar to the exclusion process of classical systems, and indeed it has been named the *quantum exclusion process* (we will discuss bosons in a separate paper). For simplicity we restrict here our discussion of duality to the boundary-driven setting. We believe our construction of a dual process extends to other quantum models as well, particularly those of exactly solvable quantum Liouvillians (or Lindbladians), beyond the quantum symmetric exclusion (e.g. [27–30]).

In a quantum stochastic systems two sources of randomness coexist: *quantum fluctuations* (due to the quantum evolution) and *dynamical fluctuations* (due to a noisy dynamics). These play a role similar to thermodynamic fluctuations and quenched disorder, in disordered systems.

## 2 Outline of duality technique

The duality technique may be described through the following steps:

- We start with a general graph, not necessarily (although often) a chain. Between the vertices of the graph particles or energy are transported, the amount per unit time is a stochastic process (Figure 1).

- We connect 'leads' of the graph to sources of heat or particles: the 'baths'. These may be constructed from first principles by considering each bath as a very large (in fact, infinite) equilibrium system. Transport occurs when the equilibria of the baths are incompatible (Figure 2).

- In all systems where duality has been introduced, it happens that the Hamiltonian generating the dynamics may be written in terms of the generators of a (non-abelian, Lie)

group. The bulk is invariant under the group operations, while the baths are not. Acting with the group hence only modifies the baths' properties.

- Like in any stochastic system, we may compute expectation values of observables. We may then switch to the 'adjoint', in which one evolves the observables rather than the state of the system. (cf. going from Schrödinger to Heisenberg pictures).

- Crucially: *there is a judicious combination of group operation and passing to the adjoint that maps the system into a system with purely absorbing baths, and only a few particles.* This is the 'dual' setting.

- Stationary results for the original system are retrieved from the long-time results, when the baths have emptied completely the system, by counting how many particles (or how much energy) was absorbed by each lead.

We shall outline in what follows how these steps are implemented in the quantum case.

## 3 A quantum transport model

On a graph $G = (V, E)$ with vertex set $V$ and edge set $E$, we consider a set of fermionic creation/annihilation operators and the random Hamiltonian

$$H_\eta(t) = \sum_{(k,\ell) \in E} \sqrt{c_{k\ell}} \left[ a_k^\dagger a_\ell \, \eta_{k\ell}(t) + a_\ell^\dagger a_k \, \bar{\eta}_{k\ell}(t) \right], \tag{1}$$

where $\{a_i, a_j\} = 0$, $\{a_i^\dagger, a_j^\dagger\} = 0$, $\{a_i, a_j^\dagger\} = \delta_{i,j}$. This describes a system of free fermions jumping on the graph $G$, the $c_{kl}$ are coupling constants. The jump terms are noisy: the external quenched noise is given by pairs of independent and identical distributed complex conjugated Gaussian white noise (one pair for each edge of the graph) with covariances

$$\mathbb{E}[\eta_{k\ell}(t)\bar{\eta}_{k'\ell'}(t')] = \delta(t - t')\delta_{k\ell,k'\ell'}. \tag{2}$$

Using the Trotter's product formula, it then follows for the density matrix

$$\rho_\eta(t) = T\left[ \exp\left\{ -i \int_{t_0}^{t} [H_\eta(t'), \cdot] dt' \right\} \right] \rho(0), \tag{3}$$

where $T$ denotes "time-order". Expanding we have

$$\rho_\eta(t) = T\left[ \left( 1 - i \int_{t_0}^{t} dt_1 [H_\eta(t_1), \cdot] - \frac{1}{2} \int_{t_0}^{t} dt_1 \int_{t_0}^{t} dt_2 [H_\eta(t_2), [H_\eta(t_1), \cdot]] + \ldots \right) \right] \rho(0). \tag{4}$$

We are interested in the noise-dependent expectation

$$\langle A(t) \rangle_\eta = \text{Tr}[A\rho_\eta(t)], \tag{5}$$

where $A$ is a generic operator (which may be expressed in terms of $a_i$ and $a_i^\dagger$) as well as the *averaged-quenched* expectation

$$\mathbb{E}[\langle A(t) \rangle_\eta] = \text{Tr}[A\rho(t)], \tag{6}$$

where we have defined the quenched-averaged density matrix

$$\rho(t) = \mathbb{E}[\rho_\eta(t)]. \tag{7}$$

In the formulas above 'Tr' denotes the trace operation yielding the quantum expectations. Averaging (4) with respect to the external noise and using the covariances (2) we get the evolution equation [1]

$$\frac{d}{dt}\rho = -\frac{1}{2} \sum_{(k,\ell) \in E} c_{kl} \left( [a_k^\dagger a_\ell, [a_\ell^\dagger a_k, \rho]] + [a_\ell^\dagger a_k, [a_k^\dagger a_\ell, \rho]] \right)$$
$$\equiv -\mathcal{H}(\rho). \tag{8}$$

Developing the commutators $\mathcal{H}$ may be written as

$$\mathcal{H} = -\sum_{(k,\ell) \in E} c_{kl} \left( \mathcal{J}_k^+ \mathcal{J}_\ell^- + \mathcal{J}_k^- \mathcal{J}_\ell^+ + 2\mathcal{C}_k^+ \mathcal{C}_\ell^+ + 2\mathcal{C}_k^- \mathcal{C}_\ell^- - \frac{1}{2} \right), \tag{9}$$

where we have defined the following on-site *superoperators* acting as follows on an *operator* $A$:

$$
\begin{aligned}
\mathcal{J}_i^+(A) &= a_i^\dagger A a_i, \\
\mathcal{J}_i^-(A) &= a_i A a_i^\dagger, \\
\mathcal{C}_i^+(A) &= \frac{1}{2} \left( a_i^\dagger a_i A - A a_i^\dagger a_i \right), \\
\mathcal{C}_i^-(A) &= \frac{1}{2} \left( a_i^\dagger a_i A + A a_i^\dagger a_i - A \right).
\end{aligned}
\tag{10}
$$

Just by applying two of these in succession (in both orders), one can directly check they satisfy (on each vertex $i$) an $\mathfrak{u}(2)$ algebra, decomposable as $\mathfrak{su}(2)$

$$[\mathcal{C}_i^-, \mathcal{J}_i^\pm] = \pm \mathcal{J}_i^\pm, \qquad [\mathcal{J}_i^+, \mathcal{J}_i^-] = 2\mathcal{C}_i^-, \tag{11}$$

and $\mathcal{C}_i^+$, which commutes with everything $[\mathcal{C}_i^+, \mathcal{J}_i^\pm] = 0 = [\mathcal{C}_i^+, \mathcal{C}_i^-]$ and are hence constants of motion for every $i$. It is easy to see that they count the number of creation minus the number of destruction operators in every site. Throughout this paper, this number is zero: there is always an equal number of creation and destruction operators in all sites. Other representations are of course possible.

We get, for the averaged-quenched expectation value of an operator $A$ at time $t$:

$$\mathbb{E}[\langle A(t) \rangle_\eta] = \mathrm{Tr}\left[ A e^{-t\mathcal{H}} \rho(0) \right] = \mathrm{Tr}\left[ \rho^\dagger(0) e^{-t\mathcal{H}^\dagger} A^\dagger \right]^*, \tag{12}$$

where we have introduced the adjoint $\mathcal{D}^\dagger$ of a superoperator $\mathcal{D}$ as:

$$\mathrm{Tr}[A^\dagger \mathcal{D}(B)]^* = \mathrm{Tr}[B^\dagger \mathcal{D}^\dagger(A)], \qquad \forall A, B. \tag{13}$$

Note that the same symbol is used both for the adjoint $A^\dagger$ of an operator $A$ on the Hilbert space, and for the adjoint of superoperators.

Using the cyclic property of the trace, it is easy to see that:

$$[\mathcal{J}_i^\pm]^\dagger = \mathcal{J}_i^\mp \qquad \text{and} \qquad [\mathcal{C}_i^\pm]^\dagger = \mathcal{C}_i^\pm. \tag{14}$$

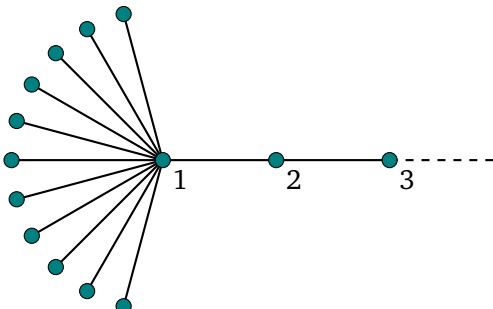

Figure 2: Schematic picture of a reservoir connected to site 1 of the graph. The bath is made of infinitely many vertices in a papyrus fashion (in the figure we represent some of them as the points connected from the left to site 1) with the same rule of transport of the bulk system. Each bath is an equilibrium system.

# 4 Real replicas and expectation values

What we have done up to now is not enough [20, 21]. Indeed, suppose we wish to calculate the following expectation: $\mathbb{E}\left[\langle A(t)\rangle_\eta \langle B(t)\rangle_\eta\right] = \mathbb{E}\left\{\text{Tr}[A\rho_\eta(t)]\text{Tr}[B\rho_\eta(t)]\right\}$. A way to do this is to replicate the system twice (all superoperators, operators and states) and use the fact that the trace of a tensor product is the product of the traces:

$$\mathbb{E}\left[\langle A(t)\rangle_\eta \langle B(t)\rangle_\eta\right] = \mathbb{E}\left\{\text{Tr}[A^\alpha B^\beta \rho_\eta^{(\alpha)} \otimes \rho_\eta^{(\beta)}(t)]\right\}, \tag{15}$$

where the 'replica index' $\alpha, \beta = 1, 2$. This will lead us to product density functions $\rho_2 = \mathbb{E}[\rho_\eta^{(1)} \otimes \rho_\eta^{(2)}]$ of two real replicas, and an averaged operator $\mathcal{H}_2(a_{i,\alpha}^\dagger, a_{i,\alpha}, a_{i,\beta}^\dagger, a_{i,\beta})$ acting on it. The noise is the same for both replicas. Note the close analogy with disordered systems: here the fermions are the replicated variables (like the spins of a spin-glass), and the noise is the disorder, playing the role of the disordered interactions $J_{ij}$. For simplicity, we shall run the steps in detail for a single replica, and then present in a more compact way the same steps for the $n$-times replicated case.

# 5 Explicit construction of a bath

We shall construct the bath as an ensemble of $s$ independent sites represented by fermionic operators $b_m^\dagger, b_m$ connected to site number 1 in a papyrus fashion like in Figure 2. The bath is larger than any other number in the problem ($s \to \infty$), and its particle density is fixed on average by its initial density matrix:

$$\rho_{bath} = \frac{e^{-\mu \sum_{m=1}^s b_m^\dagger b_m}}{(1 + e^{-\mu})^s}, \tag{16}$$

which we should tensor-product with the initial one of the chain.

Thus we consider a sum of $s$ interaction terms, which we shall assume are of intensity $c_{1,m} = \Gamma_1/s$, with $\Gamma_1 > 0$ the coupling constant to the reservoir. Denoting $\mathcal{K}^\pm \equiv (\sum_{m \in bath} \mathcal{J}_m^\pm)/s$ and $\mathcal{D}^\pm \equiv (\sum_{m \in bath} \mathcal{C}_m^\pm)/s$, the combined jump superoperators to all the leaves in the bath (which obviously obey the same algebra), we may write:

$$\mathcal{H}_1^{(s)} = -\Gamma_1 \left( \mathcal{K}^+ \mathcal{J}_1^- + \mathcal{K}^- \mathcal{J}_1^+ + 2\mathcal{D}^+ \mathcal{C}_1^+ + 2\mathcal{D}^- \mathcal{C}_1^- - \frac{1}{2} \right). \tag{17}$$

Now, let us consider the action of $\mathcal{K}^{\pm}$, $\mathcal{D}^{\pm}$ on $\rho_{bath}$. For example,

$$\mathcal{K}^+(\rho_{bath}) = \frac{1}{s}\sum_m b_m^{\dagger}\rho_{bath}b_m = \frac{1}{s}e^{\mu}\sum_m b_m^{\dagger}b_m\,\rho_{bath}\,.$$

Because $s$ is of large, by the law of large numbers we may substitute the number of fermions in the bath by its average: $\frac{1}{s}\sum_{m=1}^{s}b_m^{\dagger}b_m \to \frac{e^{-\mu}}{1+e^{-\mu}} \equiv \lambda$ as $s \to \infty$. This implies that we may substitute, for large $s$

$$\mathcal{K}^+ \to 1-\lambda\,, \qquad \mathcal{K}^- \to \lambda\,, \tag{18}$$

$$\mathcal{D}^- \to \frac{2\lambda-1}{2}\,, \qquad \mathcal{D}^+ \to 0\,. $$

The net effect is that (now allowing for several leads $i$):

$$\mathcal{H}_{bath} = -\sum_{i\in V}\Gamma_i\left[\lambda_i\left(\mathcal{J}_i^+ + \mathcal{C}_i^- - \frac{1}{2}\right) + (1-\lambda_i)\left(\mathcal{J}_i^- - \mathcal{C}_i^- - \frac{1}{2}\right)\right], \tag{19}$$

which is clearly of the Lindbladian form.

## 6 Transformations

Bearing the group structure in mind, the transformations leading to a dual process are now readable from the analogous ones in the classical system [19]. To simplify we start considering just one lead to a bath $i = 1$

$$(\mathcal{H}_1)^{\dagger} = -\left[\lambda(\mathcal{J}_1^- + \mathcal{C}_1^- - \frac{1}{2}) + (1-\lambda)(\mathcal{J}_1^+ - \mathcal{C}_1^- - \frac{1}{2})\right], \tag{20}$$

where we have used (14). Now conjugate with $e^{\mathcal{J}_1^+}$ and find

$$e^{-\mathcal{J}_1^+}(\mathcal{H}_1)^{\dagger}e^{\mathcal{J}_1^+} = -\lambda\mathcal{J}_1^- + \mathcal{C}_1^- + \frac{1}{2}\,.$$

We now consider an extended system and write $\mathcal{H} = \mathcal{H}_{bulk} + \mathcal{H}_{baths}$. Doing the same with every lead $i$, and using that the bulk superoperator $\mathcal{H}_{bulk}$ commutes with $\mathcal{J}_{tot}^+ = \sum_{i\in V}\mathcal{J}_i^+$ we get

$$\mathcal{H}^{\dagger} = e^{\mathcal{J}_{tot}^+}\left(\mathcal{H}_{bulk} - \sum_{i\in V}\Gamma_i\left\{\lambda_i\mathcal{J}_i^- - \mathcal{C}_i^- - \frac{1}{2}\right\}\right)e^{-\mathcal{J}_{tot}^+}$$
$$= e^{\mathcal{J}_{tot}^+}\mathcal{H}'e^{-\mathcal{J}_{tot}^+}\,. \tag{21}$$

Let us now reinstate baths, undoing the step we did before. We thus attach to each vertex $i \in V$ an extra site with superoperators $\mathcal{K}_i^{\pm}$ and $\mathcal{D}_i^{\pm}$ describing now empty baths (i.e. all having $\lambda = 0$, cf. Eq. (18))

$$\lambda_i\mathcal{J}_i^- - \mathcal{C}_i^- - \frac{1}{2} \to \lambda_i\mathcal{K}_i^+\mathcal{J}_i^- + 2\mathcal{D}_i^-\mathcal{C}_i^- - \frac{1}{2}\,. \tag{22}$$

Comparing to (17) we notice that the terms $\mathcal{K}_i^-\mathcal{J}_i^+$ and $\mathcal{D}_i^+\mathcal{C}_i^-$ are absent because the baths are now empty. We stress once more that, although labeled with the indexes $i$ of the vertices of the graphs, the superoperators describing the empty bath live in an additional extra space (i.e. they act on vertices attached to the bath). Finally another conjugation allows to eliminate

the disturbing factors $\lambda_i$. Indeed, using the fact that $e^{\ln \lambda_i (\mathcal{D}_i^- + 1/2)} \mathcal{K}_i^+ e^{-\ln \lambda_i (\mathcal{D}_i^- + 1/2)} = \lambda_i \mathcal{K}_i^+$ we may write

$$\mathcal{H}' = e^{\sum_i \ln \lambda_i (\mathcal{D}_i^- + 1/2)} \mathcal{H}^{dual} e^{-\sum_i \ln \lambda_i (\mathcal{D}_i^- + 1/2)}, \tag{23}$$

where

$$\mathcal{H}^{dual} \equiv \mathcal{H}_{bulk} - \sum_{i \in V} \Gamma_i \left\{ \mathcal{K}_i^+ \mathcal{J}_i^- + 2\mathcal{D}_i^- \mathcal{C}_i^- - \frac{1}{2} \right\}. \tag{24}$$

All in all, the conjugation in (21) can thus be used to write the expectation of an observable $O$ as:

$$\begin{aligned} \text{Tr}\left[ O e^{-t\mathcal{H}} \rho(0) \right] &= \text{Tr}\left[ \rho(0)^\dagger e^{-t\mathcal{H}^\dagger} O^\dagger \right]^* \\ &= \text{Tr}\left[ \rho(0)^\dagger e^{\mathcal{J}_{tot}^+} e^{-t\mathcal{H}'} e^{-\mathcal{J}_{tot}^+} O^\dagger \right]^*. \end{aligned} \tag{25}$$

As we shall later see, it is convenient to define

$$\hat{O} \equiv e^{-\mathcal{J}_{tot}^+} O^\dagger = \Pi_i e^{-\mathcal{J}_i^+} O^\dagger. \tag{26}$$

Note that because of the product form, $\hat{O}$ depends exclusively on the same sites as $O$. We have to make a choice of how we extend $O$ in the product space. We shall choose (without loss of generality) that it is $\tilde{O} = \hat{O} \otimes |0_b\rangle\langle 0_b|$: the bath sites are completely empty at time zero. Instead, the operator $\rho(0)$ is now understood as $\rho(0) \otimes 1_b$, it acts on the bath sites as the identity. Suppose for example that $O = a_k^\dagger a_k$ for a chain of length $N$, then

$$\begin{aligned} \Pi_i e^{-\mathcal{J}_i^+} (a_k^\dagger a_k)^\dagger &= (1 - a_1^\dagger a_1)...a_k^\dagger a_k...(1 - a_N^\dagger a_N) \\ &= |0\rangle\langle 0|_1... \otimes |1\rangle\langle 1|_k \otimes ... \otimes |0\rangle\langle 0|_N, \end{aligned} \tag{27}$$

i.e. a chain with one fermion in site $k$, and otherwise empty.

Going back to (25) and using once again the adjoint in (14), we get

$$\text{Tr}\left[ O e^{-t\mathcal{H}} \rho(0) \right] = \text{Tr}\left[ \left( e^{\mathcal{J}_{tot}^-} (\rho(0)) \right)^\dagger e^{-t\mathcal{H}'} (\hat{O}) \right]^*, \tag{28}$$

and inserting (23) into (28) we obtain the final result

$$\mathbb{E}[\langle O(t) \rangle_\eta] = \text{Tr}_{bulk} \text{Tr}_{baths} \left[ \left\{ \left( e^{\mathcal{J}_{tot}^-} (\rho(0)) \right)^\dagger e^{\sum_i \ln \lambda_i (\mathcal{D}_i^- + 1/2)} \right\} e^{-t\mathcal{H}^{dual}} (\tilde{O}) \right]^*. \tag{29}$$

We have used the fact that $\tilde{O}$ has no particles in the bath sites, so that $(\mathcal{D}_i^- + 1/2)(\tilde{O}) = 0$ (cf. Eq. (10)). This is the duality result: to compute the expectation of the observable $O$ evolving with the *original process* and starting from the density matrix $\rho(0)$, we consider instead an initial density matrix $\tilde{O}$ which evolves through the *dual process*, and we evaluate the 'observable' $\left( e^{\mathcal{J}_{tot}^-} (\rho(0)) \right)^\dagger e^{\sum_i (\ln \lambda_i) (\mathcal{D}_i^- + 1/2)}$ at the end. Note that if $\tilde{O}$ has not the requirements of unit trace and positivity of a density matrix, it can always be brought into one that has those properties, by addition of a term proportional to the invariant measure, and multiplication by a constant (to normalize).

## 7 Billiard pocket

Duality relations show their power when considering infinite time evolution. When $t \to \infty$ the evolution voids the chain. We may thus expect that $\lim_{t \to \infty} e^{-t\mathcal{H}^{dual}} (\tilde{O}) = [\text{empty bulk}] \otimes O_\infty$,

where $O_\infty$ lives in the space of bath sites exclusively. Equation (29) becomes, then:

$$
\begin{aligned}
\lim_{t \to \infty} \mathbb{E}[\langle O(t) \rangle_\eta] &= \langle 0 | e^{\mathcal{J}_{tot}^-}(\rho(0)) | 0 \rangle_{bulk} \; \mathrm{Tr}_{baths} \left[ e^{\sum_i \ln \lambda_i \, (\mathcal{D}_i^- + 1/2)} O_\infty \right]^* \\
&= \mathrm{Tr}_{baths} \left[ e^{\sum_i \ln \lambda_i \, (\mathcal{D}_i^- + 1/2)} O_\infty \right]^*,
\end{aligned}
\tag{30}
$$

where we have used $\langle 0 | \left( e^{\mathcal{J}_{tot}^-}(\rho(0)) \right)^\dagger | 0 \rangle_{bulk} = \mathrm{Tr}_{bulk}[\rho(0)] = 1$. All the interesting information is stored in the matrix $O_\infty$. Because $O_\infty$ is a combination of bath sites that are either void or have one fermion, and denoting $n_i$ the total number of fermions in the bath $i$, we get the simple expression:

$$
\lim_{t \to \infty} \mathrm{Tr} \left[ O e^{-t\mathcal{H}} \rho(0) \right] = \sum_{\{n_i\}} c_{\{n_i\}}^{[O]} \, \Pi_i \lambda_i^{n_i},
\tag{31}
$$

where the coefficients $c_{\{n_i\}}^{[O]}$ depend on the observable $O$. We shall see a concrete example of this presently.

**Example.** Consider a liner chain of length $N$ connected at the extremes (denoted by 1 and $N$) to two reservoirs at densities $\lambda_L$ on the left, respectively $\lambda_R$ on the right. Suppose we are interested in the average number of "particles" at site $k$ in the stationary state. For this problem we now put $O_k = a_k^\dagger a_k$ (see Eq. (27)) and $i \in \{L, R\}$. We start the dual evolution from the density operator associated to the pure state with only one fermion at site $k$. In the long-time limit we will have either a fermion in the left or a fermion in the right bath, with probabilities $c_L^{[O_k]}$ and $c_R^{[O_k]} = 1 - c_L^{[O_k]}$, respectively. Since for a symmetric random walk

$$
c_L^{[O_k]} = 1 - \frac{k}{N+1}
\tag{32}
$$

we conclude from (31) the stationary linear profile

$$
\langle a_k^\dagger a_k \rangle = \lambda_L + \frac{\lambda_R - \lambda_L}{N+1} k, \qquad k \in \{1, \dots, N\}.
\tag{33}
$$

A similar result can be obtained on a generic graph $G$, replacing the $c_L^{[O_k]}$ with the harmonic function of the symmetric random walk on the graph $G$.

## 8 Correlation functions & real replicas

We introduce replicas (labeled by $1, 2, \dots$) by considering copies of the system *subject to same realization of the external noise $\eta$* and characterized by fermionic operators $a_{i,1}, a_{i,1}^\dagger, a_{i,2}, a_{i,2}^\dagger, \dots$ (in the bulk) and $b_{im,1}, b_{im,1}^\dagger, b_{im,2}, b_{im,2}^\dagger, \dots$ (in the bath).

A generic number $n$ of copies is described by the noise-dependent density operator given by the tensor product and we are interested in its average:

$$
\rho_n = \mathbb{E}[\rho_\eta^{(1)} \otimes \rho_\eta^{(2)} \otimes \cdots \otimes \rho_\eta^{(n)}],
\tag{34}
$$

where each $\rho_\eta^{(\alpha)}$ acts on $a_{i,\alpha}, a_{i,\alpha}^\dagger, b_{im,\alpha}, b_{im,\alpha}^\dagger$'s, which are coupled by the same realization of the external noise. This system evolves with the replicated noisy Hamiltonian $H_\eta(t) = \sum_{\alpha=1}^n \left( \sum_{(k,\ell) \in E} \left[ a_{k,\alpha}^\dagger a_{\ell,\alpha} \, \eta_{k\ell}(t) + a_{\ell,\alpha}^\dagger a_{k,\alpha} \, \bar{\eta}_{k\ell}(t) \right] \right)$. Developing up to second order, and averaging over the noise we get the evolution equation $\frac{d}{dt} \rho_n = -\mathcal{H}_n \rho_n$ with

$$
\mathcal{H}_n = -\sum_{\alpha,\beta=1}^n \sum_{(k,l) \in E} c_{kl} \left( \mathcal{J}_{k,\alpha\beta}^+ \mathcal{J}_{l,\beta\alpha}^- + \mathcal{J}_{k,\alpha\beta}^- \mathcal{J}_{l,\beta\alpha}^+ + 2\mathcal{C}_{k,\alpha\beta}^+ \mathcal{C}_{l,\beta\alpha}^+ + 2\mathcal{C}_{k,\alpha\beta}^- \mathcal{C}_{k,\beta\alpha}^- - \frac{\delta_{\alpha\beta}}{2} \right)
\tag{35}
$$

obtained by expanding commutators just as above. The operators are now:

$$\mathcal{J}^+_{i,\alpha\beta}(A) = a^\dagger_{i,\alpha} A a_{i,\beta}\,,$$

$$\mathcal{J}^-_{i,\alpha\beta}(A) = a_{i,\beta} A a^\dagger_{i,\alpha}\,,$$

$$\mathcal{C}^+_{i,\alpha\beta}(A) = \frac{1}{2}\left(a^\dagger_{i,\alpha} a_{i,\beta} A - A a^\dagger_{i,\alpha} a_{i,\beta}\right)\,,$$

$$\mathcal{C}^-_{i,\alpha\beta}(A) = \frac{1}{2}\left(a^\dagger_{i,\alpha} a_{i,\beta} A + A a^\dagger_{i,\alpha} a_{i,\beta} - A\delta_{\alpha\beta}\right)\,, \tag{36}$$

which satisfy a $\mathfrak{u}(2n)$ algebra. The operators $\sum_\alpha C^+_{i,\alpha\alpha}$ commute with everything and thus do not evolve, they count the total number of creators minus annihilators per site. We shall only be interested in the case they are all zero. $\mathcal{H}_n$ is a nearest-neighbor quantum $\mathfrak{su}(2n)$ chain.

We may now distinguish bath terms, and we obtain in an entirely analogous manner:

$$\mathcal{H}_{n,bath} = -\sum_{\alpha,\beta=1}^{n}\sum_{i\in V}\Gamma_i\Big[\mathcal{K}^+_{i,\alpha\beta}\mathcal{J}^-_{i,\alpha\beta} + \mathcal{K}^-_{i,\alpha\beta}\mathcal{J}^+_{i,\alpha\beta} + 2\mathcal{D}^+_{i,\alpha\beta}\mathcal{C}^+_{i,\alpha\beta} + 2\mathcal{D}^-_{i,\alpha\beta}\mathcal{C}^-_{i,\alpha\beta} - \frac{1}{2}\delta_{\alpha,\beta})\Big]$$

$$\rightarrow -\sum_{\alpha,\beta=1}^{n}\sum_{i\in V}\Gamma_i\Big[\lambda_{i,\alpha\beta}(\mathcal{J}^+_{i,\alpha\beta} + \mathcal{C}^-_{i,\alpha\beta} - \frac{\delta_{\alpha,\beta}}{2}) + (\delta_{\alpha\beta} - \lambda_{i,\alpha\beta})(\mathcal{J}^-_{i,\alpha\beta} - \mathcal{C}^-_{i,\alpha\beta} - \frac{\delta_{\alpha,\beta}}{2})\Big], \tag{37}$$

where we have substituted as before bath operators by their expectation values:

$$\mathcal{K}^+_{i,\alpha\beta} \rightarrow \delta_{\alpha\beta} - \lambda_{i,\alpha\beta}\,, \qquad \mathcal{K}^-_{i,\alpha\beta} \rightarrow \lambda_{i,\alpha\beta}\,,$$

$$\mathcal{D}^-_{i,\alpha\beta} \rightarrow \frac{2\lambda_{i,\alpha\beta} - \delta_{\alpha\beta}}{2}\,, \qquad \mathcal{D}^+_{i,\alpha\beta} \rightarrow 0\,. \tag{38}$$

The $\lambda_{i,\alpha\beta}$ define the bath. The most general replica-symmetric form is $\lambda_{i,\alpha\beta} = \lambda_i\delta_{\alpha\beta} + \lambda'_i$. Matrices with different $(\lambda_i, \lambda'_i)$ commute and may be diagonalized simultaneously for all baths by a rotation in the fermion space. We shall in fact not need to do this here for the following reason: one can easily show that for $\alpha \neq \beta$, $\mathrm{Tr}\big[a^\dagger_{i\alpha} a_{i\beta}\,\rho\big] = \mathbb{E}[\langle a^\dagger_i\rangle_\rho\langle a_i\rangle_\rho] \propto \lambda'_i$, but this expectation must vanish. Hence $\lambda'_i = 0$ and the $\lambda$ matrix is proportional to the identity. We get

$$\mathcal{H}_{n,baths} = -\sum_{i\in V}\sum_{\alpha=1}^{n}\Gamma_i\Big[\lambda_i(\mathcal{J}^+_{i,\alpha\alpha} + \mathcal{C}^-_{i,\alpha\alpha} - \frac{1}{2}) + (1-\lambda_i)(\mathcal{J}^-_{i,\alpha\alpha} - \mathcal{C}^-_{i,\alpha\alpha} - \frac{1}{2})\Big]. \tag{39}$$

The set of generators $(\mathcal{J}^\pm_{i,\alpha\alpha}, \mathcal{C}^-_{i,\alpha\alpha})$ build an $\mathfrak{su}(2)$ algebra for each $1 \leq \alpha \leq n$ and commute for different $\alpha$: from the point of view of the baths we are back to the single replica problem. We must just repeat the transformations for every replica:

$$\mathcal{H}' \rightarrow \mathcal{H}_{bulk} - \sum_i \Gamma_i \sum_\alpha \left\{\lambda_i \mathcal{J}^-_{i,\alpha\alpha} - \mathcal{C}^-_{i,\alpha\alpha} - \frac{1}{2}\right\} \tag{40}$$

and then continue introducing the empty bath as before, to get:

$$\mathcal{H}^{dual} = \mathcal{H}_{bulk} - \sum_{i\alpha}\Gamma_i\left\{\mathcal{K}^+_{i,\alpha\alpha}\mathcal{J}^-_{i,\alpha\alpha} + 2\mathcal{D}^-_{i,\alpha\alpha}\mathcal{C}^-_{i,\alpha\alpha} - \frac{1}{2}\right\}. \tag{41}$$

Because of the diagonal nature of the hopping into the bath operator, *the bath sites can only exchange creation and destruction operators of the same replica in pairs.* Mathematically, this comes from the fact that a diagonal bath respects an extra symmetry, generated by the operators $\mathcal{C}^{+tot}_{\alpha\alpha} = \sum_{i\in V}\mathcal{C}^+_{i,\alpha\alpha}$ which in fact count the total number of creators minus annihilators

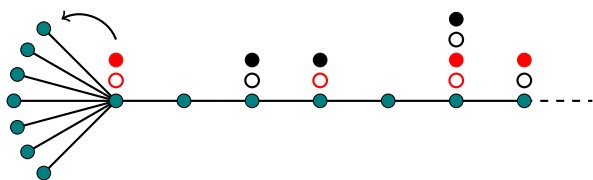

Figure 3: The configurations that are relevant in the calculation of two-point correlation functions in the test. Red and black denote replicas one and two, full points are creators and empty points are annihilators.

in the whole system for each species $\alpha$ separately. These numbers are conserved, and equal to zero for all $\alpha$ with reasonable initial conditions, and for reasonable operators $O$ with non-zero expectation. We shall restrict to this situation here. This is shown in Figure 3 where all the possibilities are shown for a two-point operator. We shall then have, for the dual chain at infinite times

$$\lim_{t\to\infty} \mathrm{Tr}\left[Oe^{-t\mathcal{H}}\rho(0)\right] = c_{\{n_{L,\alpha}\}}^{[O]} \lambda_L^{\sum_\alpha n_{L,\alpha}} + c_{\{n_{R,\alpha}\}}^{[O]} \lambda_R^{\sum_\alpha n_{R,\alpha}}, \tag{42}$$

where we have used the fact that creation and annihilation operators exit in pairs of the same species.

For example, when $O = a_{j,1}^\dagger a_{i,1} a_{i,2}^\dagger a_{j,2}$ (the case with $n = 2$ replicas), for the stationary state of a chain we have:

$$\begin{aligned}
\mathbb{E}[G_{ij}G_{ji}] &= \mathbb{E}[\mathrm{Tr}(a_{j,1}^\dagger a_{i,1} a_{i,2}^\dagger a_{j,2}\, \rho^{(1)} \otimes \rho^{(2)})] \\
&= c_{LL}^{[ij]}\lambda_L^2 + c_{RL}^{[ij]}\lambda_L\lambda_R + c_{LR}^{[ij]}\lambda_R\lambda_L + c_{RR}^{[ij]}\lambda_L^2.
\end{aligned} \tag{43}$$

Because the trace is conserved, $c_{LL}^{[ij]} + c_{RL}^{[ij]} + c_{LR}^{[ij]} + c_{RR}^{[ij]} = 0$. In particular, this means that for the stationary equilibrium $\lambda_L = \lambda_R = \lambda$ the expectation $\mathbb{E}[G_{ij}G_{ji}] = 0\ \forall i,j$. Similarly, putting $O = a_{i,1}^\dagger a_{i,1} a_{j,2}^\dagger a_{j,2}$ we have

$$\mathbb{E}[G_{ii}G_{jj}] = \mathbb{E}[\mathrm{Tr}(a_{i,1}^\dagger a_{i,1} a_{j,2}^\dagger a_{j,2}\, \rho^{(1)} \otimes \rho^{(2)}))] \tag{44}$$

and we find, at equilibrium, $\mathbb{E}[G_{ii}G_{jj}] = \lambda^2$, because the trace is now one.

In the non-equilibrium steady state with $\lambda_L \neq \lambda_R$, the expressions for the coefficients appearing in (43) have been calculated in [1]. We conclude this section by mentioning that the invariance of the super-Hamiltonian with respect to unitary rotations of the fermions in replica space may be used to find useful relations for the non-equilibrium stationary state of a chain. For example, in the case with $n = 2$ replicas, the rotation $d_i^\pm = (a_{i,1} \pm a_{i,2})/\sqrt{\pm 2}$ transforms the expectation (44) and allows us to show that:

$$\mathbb{E}[G_{ij}G_{ji}] = \mathbb{E}[G_{ii}G_{jj}]_c - \mathbb{E}[G_{iijj}]_c, \tag{45}$$

where $\mathbb{E}[G_{iijj}]_c = \langle (d_i^+)^\dagger d_i^+ (d_j^+)^\dagger d_j^+ \rangle_c = \langle n_i n_j \rangle_c^{class}$, the label 'classical' denotes the connected correlation of the classical open symmetric exclusion process (see, e.g., Eq(2.4) in [31]). This identity may be verified in the expressions of Reference [1].

## 9 Mapping the driven system into one satisfying detailed balance

Recently it was show in [19] that a large family of classical diffusive models of transport that has been considered in the past years admits a transformation into the same model in contact

with an equilibrium bath. This mapping holds for any initial conditions and is independent of the topology of the network. Such a construction provides a framework to discuss questions of time reversal in out of equilibrium contexts. Let us now briefly see that this relation holds at the quantum level with the same generality.

Consider a system in which the $\lambda_i = \lambda$ are all the same, thus making it possible to equilibrate. Transform now (40) as

$$
\begin{aligned}
\mathcal{H}'' &= e^{-\frac{\lambda}{2}J_{\alpha\alpha}^{-tot}}\mathcal{H}'e^{\frac{\lambda}{2}J_{\alpha\alpha}^{-tot}} \\
&= \mathcal{H}_{bulk} - \sum_i \Gamma_i \left\{ 2\mathcal{C}_{\alpha\alpha}^{-tot} - \frac{1}{2} \right\} = (\mathcal{H}'')^\dagger .
\end{aligned}
\tag{46}
$$

This, when $\mathcal{H}''$ is written in terms of $\mathcal{H}$, is a detailed balance property.

Now, let us go back to a general $\mathcal{H}'$ of (40) and develop the operators on which it acts in subspaces defined by the value $m$ in $\mathcal{C}_{\alpha\alpha}^{-tot}(O) = mO$. Because $[\mathcal{C}_{\alpha\alpha}^{-tot}, \mathcal{J}_{i,\alpha\alpha}^-] = -\mathcal{J}_{i,\alpha\alpha}^-$ it is clear that the terms in $\mathcal{H}'$ proportional to $\mathcal{J}_{i\alpha\alpha}^-$ (the only ones depending on the $\lambda_i$) are lower 'triangular' in this (super) matrix, do not affect the spectrum. Thus, all possible bath combinations are mappable by a similarity transformation into one another, and in particular to the situation with detailed balance (see [19,32] for an extensive discussion of the classical case).

## 10 Conclusion

We have shown how to construct a dual model for a quantum transport system. We have also shown that there is a transformation that maps all possible evolutions of the system into one in which the bath ensemble satisfies detailed balance. Because these properties are directly deduced from the group structure, the generalization to other models (e.g. the quantum Kac model [33,34] or quantum KMP [6]) should be immediate – just changing group.

## Acknowledgments

We thank Denis Bernard for helpful discussions and the anonymous referees for their comments. JK is supported by the Simons Foundation Grant No. 454943. RF is supported by the German research foundation (Deutsche Forschungsgemeinschaft DFG) Research Fellowships Programme 416527151. CG is supported by the UniMoRe project 'ALEA'.

## A Algebra and representations

The generators can be arranged into a $u(2n)$ algebra

$$
\mathcal{E}_{AB} = \begin{pmatrix} \mathcal{C}_{\alpha\beta}^+ + \mathcal{C}_{\alpha\beta}^- + \frac{\delta_{\alpha\beta}}{2} & \mathcal{J}_{\alpha\beta}^+ \\ \mathcal{J}_{\alpha\beta}^- & \mathcal{C}_{\alpha\beta}^+ - \mathcal{C}_{\alpha\beta}^- + \frac{\delta_{\alpha\beta}}{2} \end{pmatrix} ,
\tag{47}
$$

satisfying the commutation relations

$$
[\mathcal{E}_{AB}, \mathcal{E}_{CD}] = \delta_{BC}\mathcal{E}_{AD} - \delta_{DA}\mathcal{E}_{CB} .
\tag{48}
$$

The space of states consists of arbitrary combinations of the fermionic oscillators $(a_\alpha, a_\beta^\dagger)$, with $\alpha, \beta = 1, \ldots, n$. From the nilpotency of fermionic oscillators it is then clear that we are dealing with finite-dimensional representations. For a given number of replicas $n$ the space of

states can be decomposed into irreducible representations on which the first Casimir $\sum_\alpha \mathcal{C}^+_{\alpha\alpha}$, which counts the number of creation minus annihilation operators, is proportional to the identity. These reducible representations are the antisymmetric (fundamental) representations of dimension $\binom{2n}{k}$ with Dynkin weights

$$\lambda^{[k]} = (\underbrace{1,\ldots,1}_{k}, \underbrace{0,\ldots,0}_{2n-k}), \tag{49}$$

with $k = 0, 1, \ldots, 2n$. The corresponding highest weight states are

$$v^{[k]}_{hws} = \begin{cases} \prod_{i=1}^{k} a_i^\dagger \prod_{j=1}^{n} a_j, & 1 \le k \le n \\ \prod_{i=1}^{n} a_i^\dagger \prod_{j=k-n+1}^{n} a_j, & n < k \le 2n \end{cases}. \tag{50}$$

We verify that $\mathcal{E}_{AB} v^{[k]}_{hws} = 0$ for $1 \le A < B \le 2n$ and $\mathcal{E}_{AA} v^{[k]}_{hws} = \lambda^{[k]}_A v^{[k]}_{hws}$ and note that the number of creation minus annihilation operators is related to the index $k$ in (49) via $k - n$.

The Hamiltonian density, cf. (35), is then mapped to the coproduct of the second Casimir

$$\mathcal{H}_n = -\sum_{A,B=1}^{2n} \mathcal{E}^1_{AB} \mathcal{E}^2_{BA} + \frac{1}{2} \left( \sum_{A=1}^{2n} \mathcal{E}^1_{AA} + \sum_{A=1}^{2n} \mathcal{E}^2_{AA} \right). \tag{51}$$

The first Casimir $\sum_{A=1}^{2n} \mathcal{E}_{AA}$ is local and proportional to the identity for a given irreducible representation.

From the representation theory it is clear that the quantum model can be realised "classically" where $\mathcal{E}_{AB}$ are matrices acting on a vector space. The matrices are block diagonal with block sizes corresponding to the irreducible representations $\lambda^{[k]}$ where $k = 0, 1, \ldots, 2n$. Fixing the first Casimir we can restrict to the corresponding block. As described in the main text the boundary terms can be understood in the same framework. The algebra reduces to $\mathfrak{u}(2)$. It further becomes clear that there is a basis where $\mathcal{E}_{AA}$ is diagonal, $\mathcal{E}_{AB}$ with $A < B$ is upper triangular and $\mathcal{E}_{AB}$ with $A > B$ is lower triangular. This is essential for the proposed duality. We further remark that algebraically the models with $k = 1$ are equivalent to the standard multi-color SSEP, cf. [35], where the generators are replaced by the elementary matrices $\mathcal{E}_{AB} \to E_{AB}$. In the example above, see Figure 3, we have $n = 2$ and $k = 2$ such that the representation is of dimension 6. The matrix realisation of the super operator $\mathcal{H}$ can be obtained by writing the generators in the basis

$$\begin{aligned} v_1 &= 1, & v_2 &= a_1^\dagger a_1, & v_3 &= a_1^\dagger a_2, \\ v_4 &= a_2^\dagger a_1, & v_5 &= a_2^\dagger a_2, & v_6 &= a_1^\dagger a_1 a_2^\dagger a_2. \end{aligned}$$

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
