# Peer review of "Duality in quantum transport models"

_SciPost Physics, doi:SciPost Phys. 10, 135 (2021)_

## Round 1 · Referee Report · Anonymous (Referee 1) · 2020-11-19

Strengths

1- Original work on a timely subject.
2- Exact results about a potentially powerful tool.

Weaknesses

1- The overall formatting and writing can be improved.
2- The main results and examples are not worked out in enough depth which makes it difficult to see the whole scope of the approach proposed.

Report

The "duality approach" is a formal mapping which has encountered great success in the study of classical stochastic models of transport such as exclusion processes. The authors propose here to extend this tool to the quantum realm and give an explicit and exact construction of the dual theory for a recently introduced quantum stochastic model of transport, known as the QSSEP.
These results are interesting and potentially powerful and deserve to be published. In my opinion however, both the writing and the formatting of the manuscript can still be improved and the derivations done on the QSSEP could be discussed in more details -see the requested changes.

Requested changes

I list below the changes requested not in order of importance but in the order they appear in the text :

0- The abstract appears in a funny way in SciPost.

1- "In all systems where duality has been introduced, it happens that the dynamics may be written in terms of the generators of a group" :
By definition in quantum physics, the Hamiltonian belongs to the lie algebra and generates the evolution. Can the authors be a bit more precise about what they mean here?

2- Eq (10) :
The $\dagger$ symbol both stands for the adjoint of endomorphism on the Hilbert space and for the adjoint of superoperators which leads to some confusion.

3- Eq(20) :
$\tilde{\lambda_i}$ is not defined. Also I think this operation should be detailed a bit more. For instance, as far as my understanding goes, $i$ should be the index of the vertex belonging to the system and $\mathcal{K^+}$ and $\mathcal{D^-}$ should be superoperators acting on the vertices belonging to the bath (as in eq(15)).

4- In-line equation before (21) :
It seems that the part in the exponential that are just numbers should cancel with each other, what's the point in writing them? Also the cancellation used after equation (27) should rather be with $\mathcal{D}^-$ instead of $\mathcal{D}^--1$?

5- From this point on, the authors invert the role of operators and density matrices. However one can not be trivially taken for the other since the density matrix obviously fulfills more constraint such as preservation of the trace. It would be nice if this exchange was explained a bit more.

6- First paragraph in VII :
The position in the tensor product of the bath and the bulk have been inverted with respect to the previous convention which can lead to confusion.

7- Eq (29) :
The step from (28) to (29) is a bit abrupt, especially since it is one of the important results of the paper. It would be good to give a more detailed explanation on how the coefficients $c^{[O]}_{\{n_i\}}$ are related to $O$.

8- In the same manner, it would be great to have more details on how the results (30), (31), (41) have been derived, as they constitute important illustrations on how the formalism works.

9- Paragraph before (42) :
What is the result for $\lambda_L \neq \lambda _R$?

10- In general it is not clear from the results presented how this formalism helps getting more results than those which were presented in [4]. Can the authors comment on that?

11-"the generalization to other models (e.g. the quantum Kac model[8, 20] or quantum KMP [21]) should be immediate –just changing group."
In the conclusion, the authors state that the formalism presented can be easily applied to other type of models. It would be nice to have a more precise statement on the general requirements for a given quantum model to have a dual structure. For instance, from the construction, it seems to me that the algebra describing the generators must fulfill some strict conditions.

---

## Round 1 · Referee Report · Anonymous (Referee 2) · 2020-12-11

Strengths

1- Extension of stochastic duality to quantum processes

2- Mapping to equilibrium

Weaknesses

1- Some useful explanations may be added.

Report

\documentclass[12pt]{article} \usepackage{graphicx}
\usepackage{amsmath} \usepackage{amssymb} \usepackage{amscd}

\begin{document}

\centerline{\bf Referee Report: Duality in quantum transport models,}
$\,\,\quad\quad$ {\it by R. Frassek, C. Giardina and J. Kurchan}

\vskip 0.3cm

The authors develop the duality concept for quantum transport models
by constructing explicitly a Hamiltonian dual to the Hamiltonian of
free fermions on a graph with external quenched noise. Thanks to
this duality, the (quantum) expectation of an observable can be
calculated by considering a finite number of particles. This
manuscript extends to the quantum realm a technique that has been
become essential for classical interacting particles. As thus the
present work is likely to be a seminal contribution in the field of
non-equilibrium quantum dynamics.

\vskip 0.3cm

The manuscript is written in with a genuine pedagogical intent. The
authors build on their knowledge of classical duality to show how
each step can be transposed to the quantum case. They provide enough
details in general to allow the reader to reproduce most of the
calculations and to convey to the reader the importance of their
results. However, it may be useful at some places to provide to the
reader with some supplementary explanations, rationale, or
extra-references (see below).

\vskip 0.3cm

Here are some minor remarks on the manuscript:

\vskip 0.1cm

1) In the long introduction, background and motivations are given for
duality. However, some extra-references could be useful. In the
context of classical stochastic processes, duality has been discussed
by G. M. Sch\"utz 1997, Duality Relations for Asymmetric Exclusion
Processes, J. Stat. Phys., 86, 1265–1287 and more recently by
T. Imamura and T. Sasamoto 2011, Current moments of 1D ASEP by
duality, J. Stat. Phys 142, 919-93. It has then become an essential
tool in the field of Integrable Probabilities, see for example
A. Borodin, I. Corwin, and T. Sasamoto 2014, From duality to
determinants for q-TASEP and ASEP, Ann. Prob., 42, 2314–2382 (or any
review paper or lecture notes by these authors or their
collaborators).

\vskip 0.1cm

2) Similarly, in the introduction, the authors could give more
examples of exactly solvable quantum Liouvillians (or Lindbladians),
beyond the quantum symmetric exclusion. (Maybe, one could refer
to some works by the groups of Essler, Prosen, Caux, Calabrese
...).

\vskip 0.1cm

3) It would be helpful to give some details or references for the
derivation of Eq. (6).

\vskip 0.1cm

4) Similarly, a rationale for finding the su(2) algebra (8) and (9)
could be given.

\vskip 0.1cm

5) Section V, below eq. (14), $c_{1,m}$ should probably be replaced
by $\sqrt{c_{1,m}}$.

\vskip 0.1cm

6) The authors obtain a Lindbladian in Eq. (17). Usually this form
requires a Markovian assumption, weak coupling and a small time
expansion. Would it be possible to trace down these assumptions in
the present context?

\vskip 0.1cm

7) Between Eq. (19) and Eq. (20) the symbol $\tilde\lambda_i$ is
used. I have not been able to find the definition of this quantity in
the text before. In fact, I do not understand what the authors mean
in Eq. (20) ; some details will be useful (the deduction of the
duality Eq. (27) then follows from pure algebra).

\vskip 0.1cm

8) In Eq. (27), $\rho_o$ should be replaced by $\rho(0)$.

\vskip 0.1cm

9) I would suggest to move section IV (real replicas) just before
section VII.

\vskip 0.1cm

10) When applying duality to the quantum exclusion process (Eq. (43))
the authors may describe in more details that model.

\vskip 0.1cm

11) Section IX contains a very important result and is deceptively
short. It may even be overlooked. I would advise that the authors
discuss in more details this mapping to an equilibrium model. This
was already remarkable in the classical case !

\vskip 0.1cm

12) The authors have considered a model of hopping free fermions. Is
it possible to construct a dual model for an interacting system? It
is known that the classical symmetric exclusion which is self-dual
is equivalent to the (Wick rotated) XXX spin chain: does the quantum
XXX model also satisfy some kind of duality?

\vskip 0.5cm

I recommend publication of this manuscript in SCiPost.
\end{document}

---

## Round 2 · Referee Report · Anonymous (Referee 3) · 2021-3-15

Report

I would like to apologize to the authors and editor for the late answer, it seems that I didn't get an email notification of the re submission and I only became aware of it recently.
The authors have addressed most of my points in a satisfactory manner and I thus find the manuscript suitable for publication in the present form.

---

## Round 2 · Author Response

Dear Editor,

we carefully read the very interesting reports of the referees and thank them for their work. We modified our manuscript accordingly. Please find the changes below.

%%%%%%%%%%%%%%%%%%%%%%%%%%%%%%%%%%%%

REFEREE 1:

(0) The abstract appears in a funny way in SciPost.

It has been now fixed.

(1) "In all systems where duality has been introduced, it happens that the dynamics may be written in terms of the generators of a group" :
By definition in quantum physics, the Hamiltonian belongs to the lie algebra and generates the evolution. Can the authors be a bit more precise about what they mean here?

We changed “the dynamics” -> “Hamiltonian generating the dynamics”

(2) Eq (10) :
The † symbol both stands for the adjoint of endomorphism on the Hilbert space and for the adjoint of superoperators which leads to some confusion.

To avoid the confusion, this is now explicitly mentioned in the text

(3) Eq(20) :
\tilde{λ}_i is not defined. Also I think this operation should be detailed a bit more. For instance, as far as my understanding goes, i should be the index of the vertex belonging to the system and K+ and D− should be superoperators acting on the vertices belonging to the bath (as in eq(15)).

Thanks. We added some text.

(4) In-line equation before (21) : It seems that the part in the exponential that are just numbers should cancel with each other, what's the point in writing them? Also the cancellation used after equation (27) should rather be with D− instead of D− − 1?

Thank you, we corrected a mistake, now it is ok.

(5) From this point on, the authors invert the role of operators and density matrices. However one can not be trivially taken for the other since the density matrix obviously fulfills more constraint such as preservation of the trace. It would be nice if this exchange was explained a bit more.

We added the following comment:

Note that if $\tilde O$ has not the requirements of unit trace and positivity of a density matrix,
it can always be brought into one that has those properties, by addition of a term proportional to the invariant
measure, and multiplication by a constant (to normalize).

(6) First paragraph in VII :
The position in the tensor product of the bath and the bulk have been inverted with respect to the previous convention which can lead to confusion.

Thanks. Now it has been restored.

(7) Eq (29) :
The step from (28) to (29) is a bit abrupt, especially since it is one of the important results of the paper. It would be good to give a more detailed explanation on how the coefficients c[O]{n_i} are related to O.

We have clarified this point with the example that follows (29).

(8) In the same manner, it would be great to have more details on how the results (30), (31), (41) have been derived, as they constitute important illustrations on how the formalism works.

See (27) (and we added in the text).
We also improved the notation in (42).

(9) Paragraph before (42) : What is the result for λL ≠λR?

We added a text explaining that the result was given in ref [4].

(10) In general it is not clear from the results presented how this formalism helps getting more results than those which were presented in [4]. Can the authors comment on that?

The duality formalism helps here as much as it does in the classical setting.
Namely, it allows to describe correlation functions by just following dual
particles, whose dynamics is qualitatively immediately guessed.
As far as the stationary state is concerned, as the referee remarks,
the solution of the open quantum exclusion process is given in [4].
We remark that, should one be able to follow the dual dynamics at any time t, then
one could compute time-dependent correlation functions.
Furthermore, in the macroscopic hydrodynamic limit, it is conceivable that particles will be effectively independent,
thus simplifying the analysis.
What we really learn from duality applied to the open quantum exclusion process
is discussed in section IX, i.e. the mapping between equilibrium and non-equilibrium
systems.

(11) “the generalization to other models (e.g. the quantum Kac model[8, 20] or quantum KMP [21]) should be immediate –just changing group."
In the conclusion, the authors state that the formalism presented can be easily applied to other type of models. It would be nice to have a more precise statement on the general requirements for a given quantum model to have a dual structure. For instance, from the construction, it seems to me that the algebra describing the generators must fulfill some strict conditions.

We agree with the referee. It should be pointed out that this
has been an open problem for many years.
This will be discussed in a forthcoming paper by the authors.

%%%%%%%%%%%%%%%%%%%%%%%%%%%%%%%

REFEREE 2:

(1) In the long introduction, background and motivations are given for
duality. However, some extra-references could be useful. In the
context of classical stochastic processes, duality has been discussed
by G. M. Sch\"utz 1997, Duality Relations for Asymmetric Exclusion
Processes, J. Stat. Phys., 86, 1265–1287 and more recently by
T. Imamura and T. Sasamoto 2011, Current moments of 1D ASEP by
duality, J. Stat. Phys 142, 919-93. It has then become an essential
tool in the field of Integrable Probabilities, see for example
A. Borodin, I. Corwin, and T. Sasamoto 2014, From duality to
determinants for q-TASEP and ASEP, Ann. Prob., 42, 2314–2382 (or any
review paper or lecture notes by these authors or their
collaborators).

All the suggested references have been added.

(2) Similarly, in the introduction, the authors could give more
examples of exactly solvable quantum Liouvillians (or Lindbladians),
beyond the quantum symmetric exclusion. (Maybe, one could refer
to some works by the groups of Essler, Prosen, Caux, Calabrese
...).

Done! We added the references and new sentence at the end of
the introduction.

(3) It would be helpful to give some details or references for the
derivation of Eq. (6).

We added a sentence explaining that Eq. (6) (which is Eq. (8) is the new version)
is just a consequence of the the fact that the external noise is Gaussian.
We also cited reference [4].

(4) Similarly, a rationale for finding the su(2) algebra (8) and (9)
could be given.

We have added a phrase about applying two superoperators in succession.

(5) Section V, below eq. (14), c_{1,m} should probably be replaced
by \sqrt{c_{1,m}}.

Done!

(6) The authors obtain a Lindbladian in Eq. (17). Usually this form
requires a Markovian assumption, weak coupling and a small time
expansion. Would it be possible to trace down these assumptions in
the present context?

The Markovian assumption comes from the assumption that the noise is white.

(7) Between Eq. (19) and Eq. (20) the symbol
\tilde{λ}_i \to 0 is used. I have not been able to find
the definition of this quantity in
the text before. In fact, I do not understand what the authors mean
in Eq. (20) ; some details will be useful (the deduction of the
duality Eq. (27) then follows from pure algebra).

We added some extra text to explain what is meant.

(8) In Eq. (27), \rho_0 should be replaced by
\rho(0).

Done!

(9) I would suggest to move section IV (real replicas) just before
section VII.

We prefer to keep the current structure: section 4 is introductory and
section 8 contains the explicit computations.

(10) When applying duality to the quantum exclusion process (Eq. (43))
the authors may describe in more details that model.

The process was defined at the very beginning of the paper (Eq.(1)).
The results of Eq.(43) refer to the one-dimensional lattice geometry,
which is now recalled in the paper.

(11) Section IX contains a very important result and is deceptively
short. It may even be overlooked. I would advise that the authors
discuss in more details this mapping to an equilibrium model. This
was already remarkable in the classical case !

We have added a paragraph.

(12) The authors have considered a model of hopping free fermions. Is
it possible to construct a dual model for an interacting system? It
is known that the classical symmetric exclusion which is self-dual
is equivalent to the (Wick rotated) XX spin chain: does the quantum
XXX model also satisfy some kind of duality?

As far as we know the answer is no.

---

## Editorial Decision

published